# Correlation and Regression Analysis of Spraying Process Quality Indicators

Beata Cieniawska [1,*] , Katarzyna Pentoś [1] and Tomasz Szulc [2]

1   Institute of Agricultural Engineering, Wrocław University of Environmental and Life Sciences, 37a Józefa Chełmońskiego Street, 51-630 Wrocław, Poland
2   Łukasiewicz Research Network—Poznań Institute of Technology, 6 Ewarysta Estkowskiego Street, 61-755 Poznań, Poland
*   Correspondence: beata.cieniawska@upwr.edu.pl; Tel.: +48-666-988-949

**Abstract:** The study presents the results of the correlation and regression of the deposition of liquid and the degree of coverage of sprayed objects. Preliminary experiments were conducted in terms of droplet size depending on liquid pressure and nozzle type. Studies on the degree of coverage and deposition of spray liquid were then carried out. The test stand consisted of a carrier of nozzles and artificial plants. Samplers were attached to the artificial plants to obtain vertical and horizontal surfaces. Water-sensitive paper and filter papers were sampled (for measurements of the degree of coverage and deposition of liquid, respectively). The results of these studies showed strong and very strong Pearson's correlation coefficients between the analyzed indicators (degree of coverage and deposition of liquid), from 0.9143 to 0.9815. Furthermore, high values of the coefficient of determination ($R^2 > 0.85$) were obtained for linear regression. The high $R^2$ values indicate a good match of the regression model to empirical data.

**Keywords:** standard nozzle; air-induction nozzle; deposition of spray liquid; degree of coverage of sprayed objects

## 1. Introduction

One of the primary practices of agricultural production is chemical plant protection because it increases the control of pests and ensures high quantity and quality of yield [1]. The expected effectiveness of spraying involves the need for an adequate amount of plant protection product and uniformity of the liquid distribution in the targeted area. However, during spraying, other phenomena such as drift of spraying, environmental pollution, and poisoning of machine operators and bystanders may occur [2–5]. The authors emphasized the need to minimize losses and maximize spraying effectiveness [6,7]. It is therefore important to ensure an appropriate degree of liquid deposition and uniformity of liquid distribution while protecting humans, animals, and the environment. Relevant parameters and working conditions for sprayers in terms of atmospheric conditions should be adapted for this purpose [8]. Among the technical and technological factors include; spraying speed, dose and liquid pressure, the height of the sprayer boom, as well as the nozzles and adjuvants used [9–14].

The evaluation of spraying quality is conducted based on three indicators. The degree of coverage of sprayed surfaces and the distribution of the spray liquid are qualitative indicators, while the deposition of the liquid is a quantitative indicator.

In the literature, the degree of coverage of sprayed surfaces is defined as the ratio of the surface covered by the liquid to the total surface of the sampler. This ratio is calculated on the basis of a computer image analysis. Water-sensitive paper is the most common sampler [15]. Alternatively, the researchers proposed a fluorescent marker and the results were obtained using ultraviolet (UV) radiation [16]. In addition, Li et al. [17] presented the use of single leaf and Matlab software for the examination of coverage degree. Many

research efforts have focused on determining the impact of factors mentioned previously on the degree of coverage of spraying surfaces [18–24]. The analysis of water-sensitive papers was used to estimate the residues of the plant protection product (pentoconazole) on apple leaves. The applied correction factor was an essential aspect of the analysis. The authors highlighted the need to continue the research to determine whether the presented correction factor would be appropriate during studies with the use of other preparations [25].

The deposition of the spray liquid is a quantitative indicator, calculated as the mass of the product retained on the protected surface. Filter papers are samplers and fluorescent dyes such as rhodamine [26], mylar cards [10], and tartrazine [27] are used during experiments.

Studies on the uniformity of the distribution of liquid precipitation are usually carried out by using a grooved table. Experiments are based on the determination of the parameters of the lateral and longitudinal distribution of liquid. The impact of technical and technological factors on this indicator was described based on the results of research [28,29].

The analysis of the values of a correlation coefficient or regression models are very common approaches in various fields and disciplines such as medicine [30,31], biology [32,33], food technology [34,35], earth sciences [36,37], economics [38,39] and agriculture [40–44]. Cerruto et al. [45] stated that the coverage degree values obtained on water-sensitive papers were correlated with the deposition of liquid spray. In addition, a model for estimating liquid deposition using the degree of coverage was presented by Cerruto et al. [46]. The research was carried out in an orchard with the use of an ATR 80 hollow-cone nozzle. A strong correlation between the analyzed indicators was achieved by the authors. Wen et al. [47] obtained a correlation coefficient of deposition and coverage of 0.89. The research was carried out on the crop of cotton with the use of unmanned aerial vehicles.

The determination of the degree of coverage is generally faster and easier to conduct. However, the complex assessment of spraying quality requires both indicators described above. Hence, the main purpose of this research is to establish the correlation and regression models between the degree of coverage and deposition of liquid spray. Biological efficacy depends on the cover and deposition of liquid on plants. The models developed in this study are of high practical use. Based on the knowledge of the degree of coverage (which is faster and easier for determination) the value of deposition of spray liquid can be estimated with high precision. The results of the study may be a reference point for studies on the evaluation of the biological efficacy of the spraying process. Future studies in this area should focus on experiments in field conditions.

## 2. Materials and Methods

### 2.1. Experimental Set

The test stand consisted of a self-propelled carrier of nozzles and artificial plants (Figure 1). Measuring device was equipped with a liquid system and driving system, which allowed the liquid pressure and spraying speed to be varied. These systems were controlled by a control panel. The determination of the driving speed was carried out using a frequency converter. The electronic stopwatch, which was turned on and off by the limit switch, was also used to determine the vehicle's driving speed. The switches were placed at the beginning and at the end of the measurement section. The driving speed was calculated from the distance traveled (15 m) and the time measured by a stopwatch. The construction of carriers of nozzles made it possible to change the height of the sprayer boom. The carrier of nozzles was moving over a 30-m-long route. The travel route of the machine was divided into three parts. In the first section, the desired speed was achieved. Measurements were carried out in the second section (15 m) and the vehicle was slowing down to a complete stop in the final section. Six artificial plants were placed in the area of the measuring section. Samplers were set to artificial plants to obtain horizontal surfaces (upper and bottom) and vertical surfaces (approach and leaving). Water-sensitive papers and filter papers were used as samplers. Three artificial plants fitted with water-sensitive paper were used to assess the degree of coverage. The evaluation of the deposition of spray

liquid was made with the use of three artificial plants with filter papers attached. The artificial plants were arranged alternately and they represented three replications of the measurements (Figure 2). The water with a fluorescent product—Brilliant Sulfo Flavine (BSF)—was used during the research.

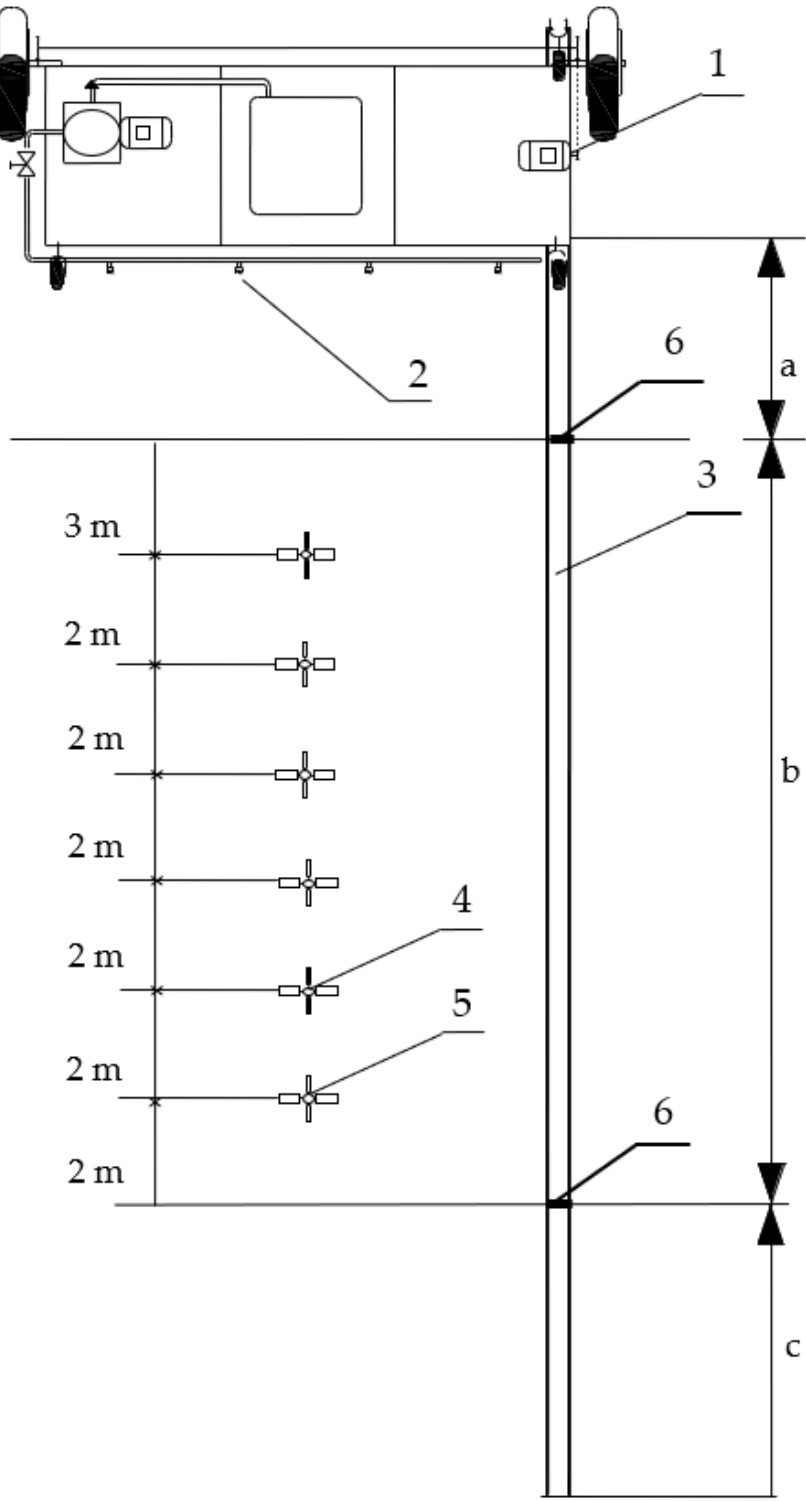

**Figure 1.** General view of measurement stand: a—inrun section, b—measurement section, c—final section, 1—sprayer, 2—nozzle, 3—guideway, 4—an artificial plant with filter paper, 5—an artificial plant with water-sensitive paper, 6—limit switch.

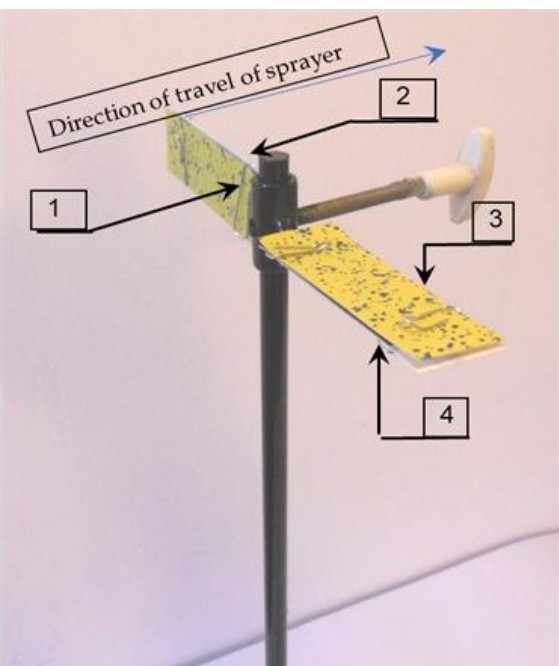

**Figure 2.** Artificial plant with water-sensitive papers: 1—vertical approach surface, 2—vertical leaving surface, 3—horizontal upper surface, 4—horizontal bottom surface.

### 2.2. Analysis of the Coverage Degree of Sprayed Surfaces

Carrier of nozzles passed over the artificial plants in measurement section. Water-sensitive papers were attached to the prepared patterns and protected against moisture. Then, water-sensitive papers were scanned. The assessment of the coverage degree was conducted in the graphic software Adobe Photoshop CC 2019. Water-sensitive papers are coated by a layer of Bromoethyl Blue, which changes in color from yellow to dark blue after contact with water. The three fragments with an area of 1 cm$^2$ were randomly selected on the surface of each sampler. The degree of coverage was calculated as a ratio of the colored surface to the area of the sample. The spraying liquid was not found on the bottom horizontal surface, thus this area has not been taken into account for further analysis.

### 2.3. Analysis of the Deposition of Spray Liquid

First, 30 mL of deionized water was added to each sample and then all samples were shaken for 15 min at a special position on the shaking frequency of 162 cycles·min$^{-1}$ and shaking amplitude of 40 mm. The assessment of the deposition of spray liquid was conducted on the luminescence fluorometer PerkinElmer LS 55.

### 2.4. Nozzle Type and Droplet Size Classification

The measurement of droplet size was carried out at Łukasiewicz Research Network—Industrial Institute of Agricultural Engineering. The measurement stand consisted of a particle spectrum analyzer (Spraytec—Malvern Panalytical Ltd, Malvern, United Kingdom) placed on the laboratory table and a guideway of the test nozzle positioned over the particle spectrum analyzer. The general view of the stand is presented in Figure 3. The number of droplets was recorded using computer software. The results of the measurements were used to determine the volume median diameter—VMD.

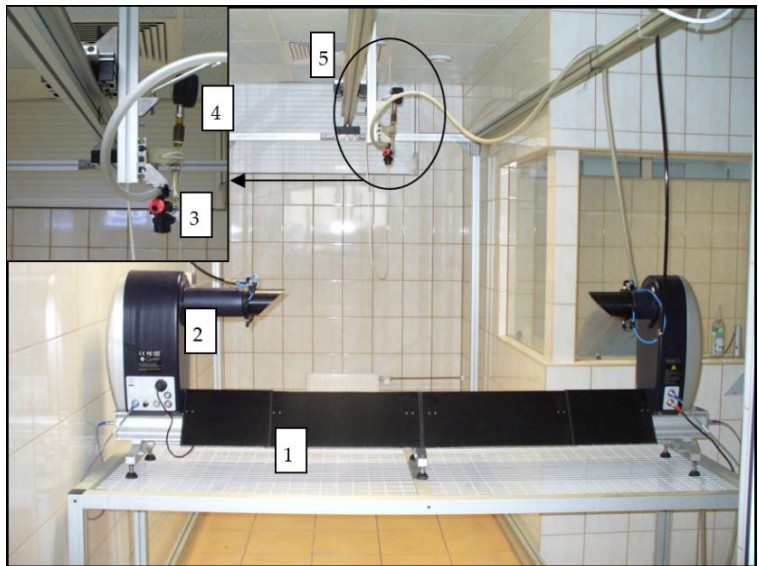

**Figure 3.** Measurement stand to the analysis of droplet size: 1—laboratory table, 2—particle spectrum analyzer, 3—nozzle body, 4—manometer, 5—guideway.

Single-stream and dual-stream nozzles were selected for the study, both standard and air induction. Droplet size classifications are based on the ASABE S572.1 [48]. The characteristics of the nozzles are detailed in Table 1.

**Table 1.** Characteristics of selected nozzles.

| Nozzle | Pressure (kPa) | Flow Rate ($dm^3 \cdot min^{-1}$) | Droplet Size (µm) | | | Drop Size Class |
|---|---|---|---|---|---|---|
| | | | $D_V 0.1$ | $D_V 0.5$ | $D_V 0.9$ | |
| DF 12002 | 200 | 0.65 | 127 | 221.1 | 332.8 | fine |
| DF 12002 | 400 | 0.91 | 110.5 | 191.6 | 275.7 | fine |
| XR 11002 | 200 | 0.65 | 105.7 | 206 | 350.1 | fine |
| XR 11002 | 400 | 0.91 | 88.8 | 178.2 | 295.3 | fine |
| CVI 11002 | 200 | 0.65 | 180.2 | 385.6 | 699 | coarse |
| CVI 11002 | 400 | 0.92 | 143 | 296.7 | 510.2 | medium |
| CVI TWIN 11002 | 200 | 0.65 | 205.1 | 468.9 | 822.7 | very coarse |
| CVI TWIN 11002 | 400 | 0.92 | 164.9 | 336.9 | 557.7 | coarse |

*2.5. Research Conditions*

The following conditions and parameters of nozzles were used for the research:

- Speed of the sprayer—2.2 $m \cdot s^{-1}$;
- Pressure of liquid—200 and 400 kPa (the highest and lowest value of the liquid pressure due to the nozzles used);
- Height of the sprayer boom—0.5 m.

*2.6. Data Processing*

All the experiments were carried out in triplicate. To evaluate the strength of linear relationships between the deposition of spray liquid and the degree of coverage, Pearson's correlation coefficient (R) was used with a *p*-value of 0.05 as statistical significance. R may take on a range of values from −1 to +1. The guidance of correlation coefficient interpretation was developed based on [49] and is presented in Table 2.

**Table 2.** The interpretation of the correlation coefficient.

| Correlation Coefficient | Interpretation |
|---|---|
| 0 | no linear relationship |
| (0; 0.40)/(−0.40; 0) | weak positive linear relationship/ weak negative linear relationship |
| <0.40; 0.70)/(−0.70; −0.40> | moderate positive linear relationship/ moderate negative linear relationship |
| <0.70; 0.90)/(−0.90; −0.70> | strong positive linear relationship/ strong negative linear relationship |
| <0.9; 1)/(−1; −0.9> | very strong positive linear relationship/ very strong negative linear relationship |
| +1/−1 | perfect positive linear relationship/ perfect negative linear relationship |

To determine relationships between indicators of spraying quality, a linear regression analysis was carried out. The coefficient of determination ($R^2$) was used to assess the fit of the linear regression model to empirical data. All statistical analyses were performed with the use of Statistica v. 13.1 software, Tibco Software, US.

## 3. Results

The results were statistically analyzed. Table 3 presents the results of the analysis of variance of the indicators of spray quality. The F test was used to evaluate the significance of the results at a significance level of $p = 0.05$.

**Table 3.** The analysis of variance.

| Nozzles | Surface | F | Se | S(a) | S(b) | Ve [%] | R |
|---|---|---|---|---|---|---|---|
| XR and DF—standard nozzles | horizontal upper | 166.4558 | 711.7962 | 12.5976 | 622.7839 | 14.7221 | 0.9398 |
| | vertical approach | 359.9703 | 113.7631 | 11.6734 | 75.0449 | 8.3603 | 0.9708 |
| | vertical leaving | 860.9794 | 87.5600 | 7.4268 | 35.1296 | 6.9483 | 0.9875 |
| CVI and CVI TWIN—air-induction nozzles | horizontal upper | 112.1109 | 817.1803 | 15.2684 | 555.8545 | 19.4904 | 0.9143 |
| | vertical approach | 130.0897 | 147.1545 | 24.3045 | 146.7021 | 11.6151 | 0.9249 |
| | vertical leaving | 578.5757 | 47.7444 | 6.0044 | 25.4283 | 4.1824 | 0.9815 |

Se—standard deviation of the residual component; S(a)—standard error of the regression parameter a; S(b)—standard error of the regression parameter b; Ve—residual variation coefficient.

The analysis of variance was divided into two parts depending on the nozzle type. XR and DF are standard nozzles, while CVI and CVI TWIN are air-induction nozzles. The analysis was performed separately for each surface. Based on the correlation coefficients (R) presented in Table 3, it can be stated that all relationships are strong or very strong with very high statistical significance ($p < 0.0005$).

The results of linear regression are shown in Figures 4–9. The values of the degree of coverage are presented on the x-axis and the values of the deposition of liquid are shown on the y-axis. The regression line, regression equation and the value of the coefficient of determination are also depicted in the plots. Figures 4–6 present the scatter plots of the degree of coverage and deposition of liquid for selected standard nozzles. The values of the deposition of liquid increase with the increase in the degree of coverage. In the case of upper horizontal surfaces, the coefficient of determination $R^2$ is higher than 0.88, which means the high quality of the regression model. While in the case of the vertical surfaces, the very high quality of the models is observed with an $R^2$ higher than 0.94.

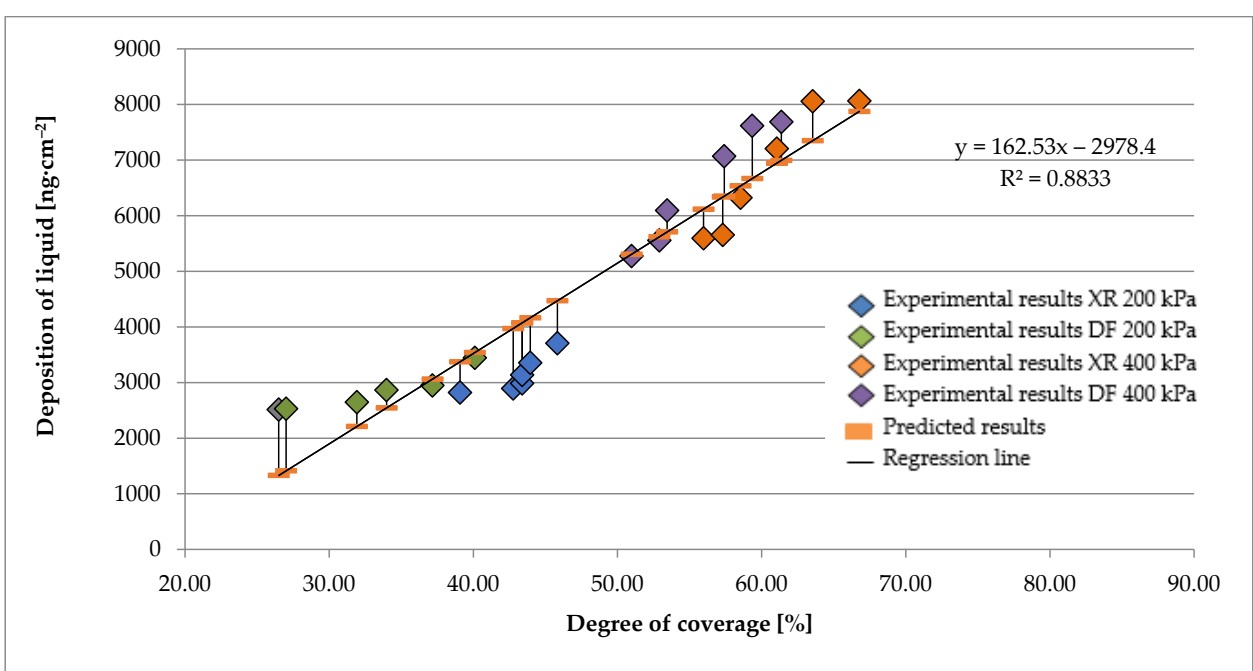

**Figure 4.** The scatter plot of the degree of coverage and deposition of spray liquid for selected standard nozzles for upper horizontal surface.

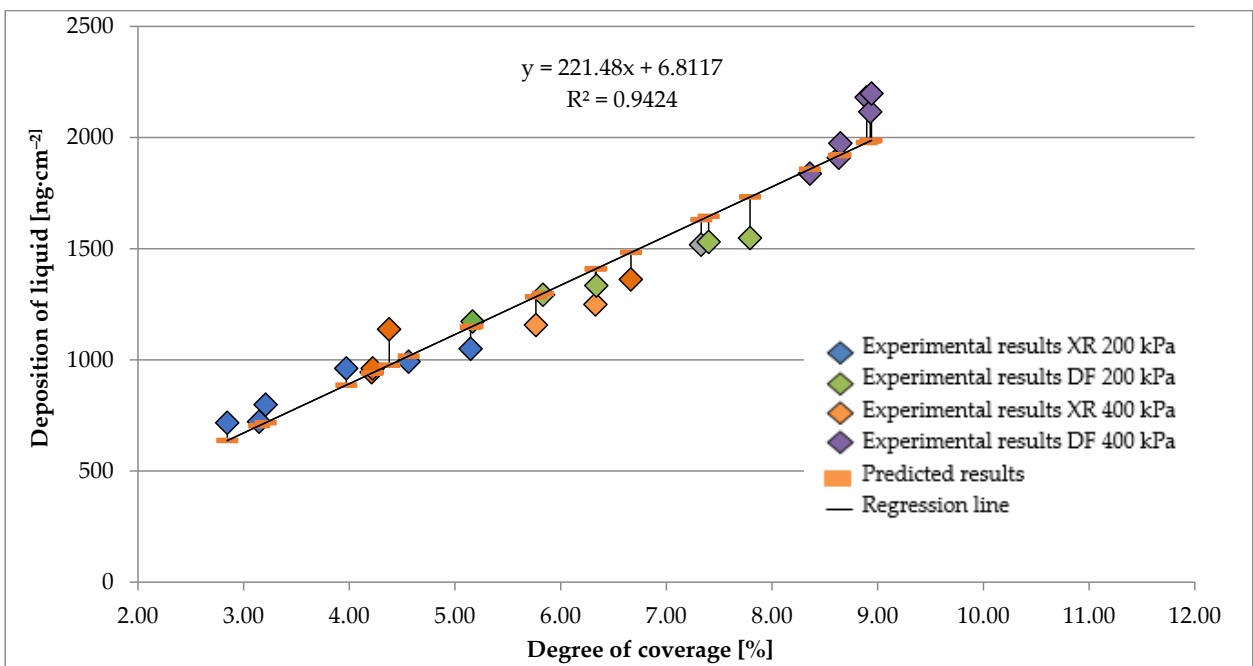

**Figure 5.** The scatter plot of the degree of coverage and deposition of spray liquid for selected standard nozzles for approach vertical surface.

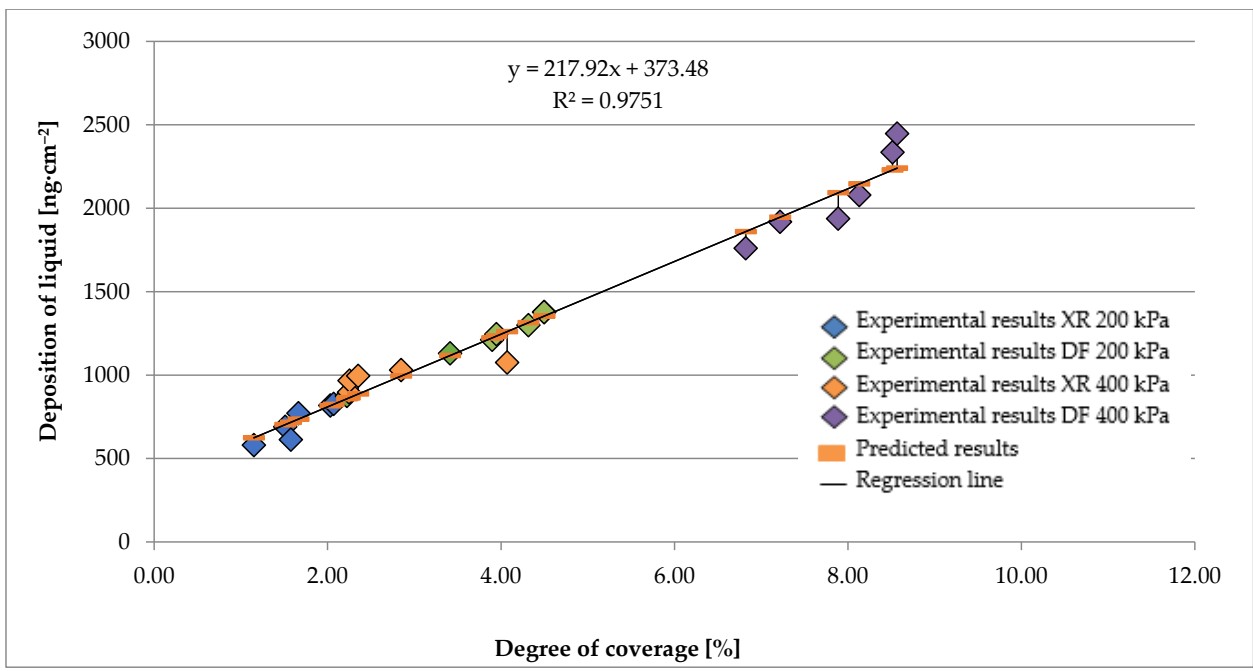

**Figure 6.** The scatter plot of the degree of coverage and deposition of spray liquid for selected standard nozzles for leaving the vertical surface.

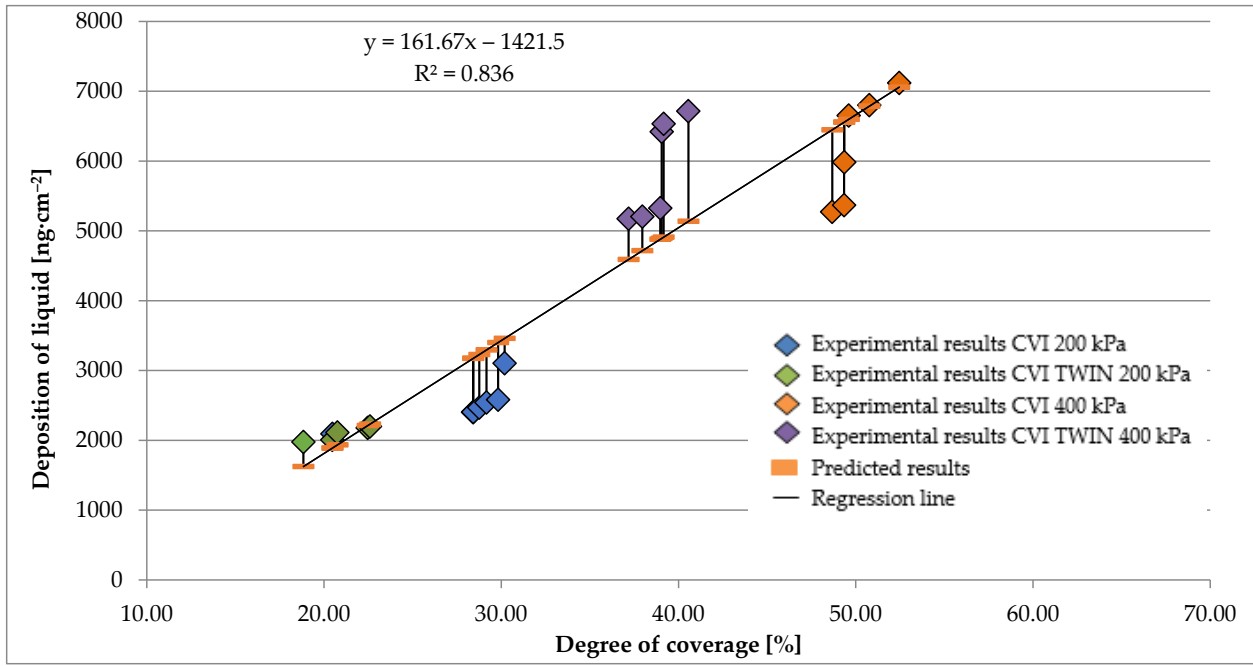

**Figure 7.** The scatter plot of the degree of coverage and deposition of spray liquid for selected air-induction nozzles for upper horizontal surface.

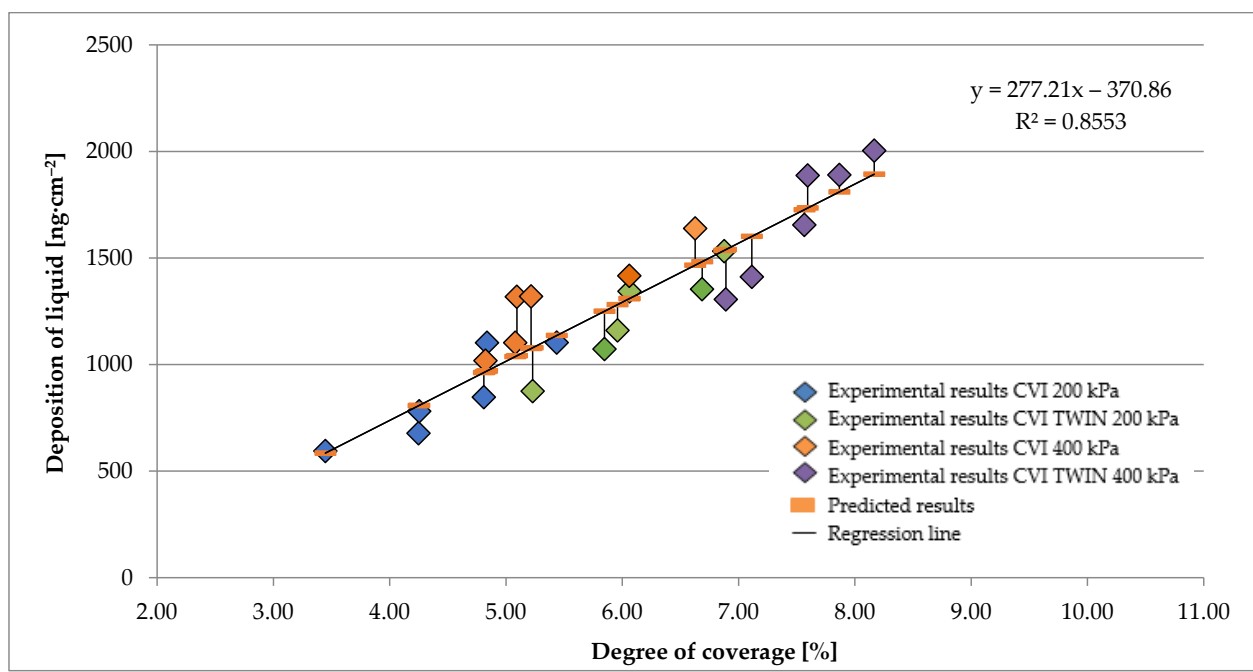

**Figure 8.** The scatter plot of the degree of coverage and deposition of spray liquid for selected air-induction nozzles for approach vertical surface.

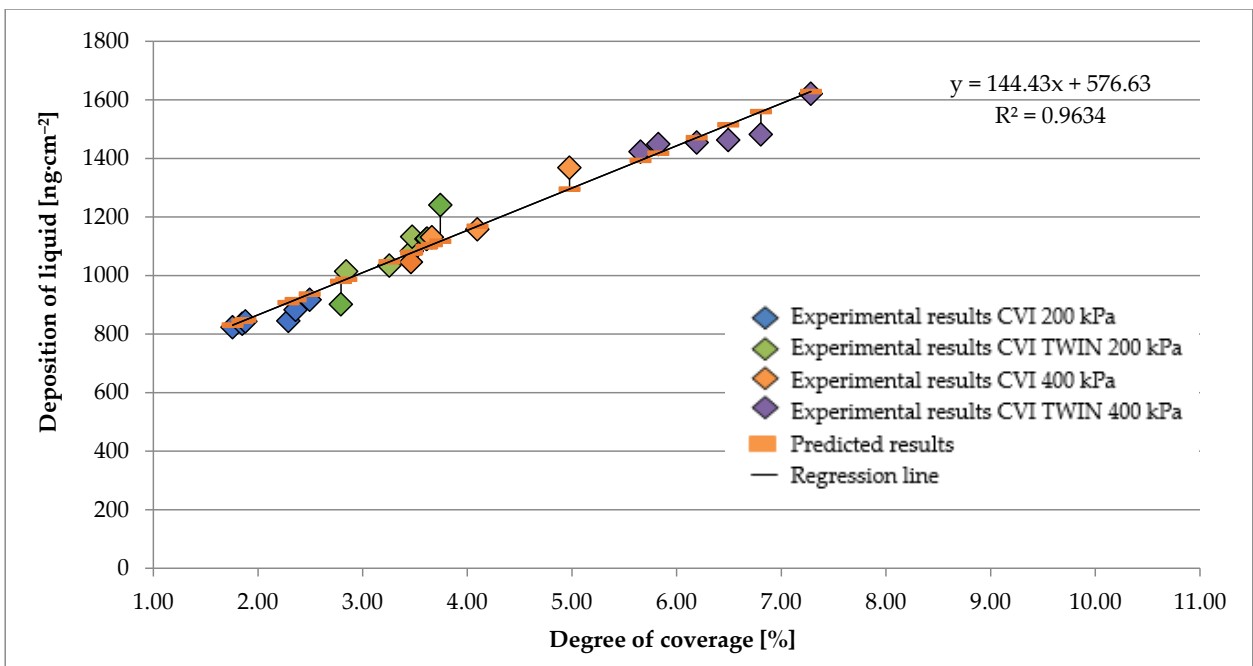

**Figure 9.** The scatter plot of the degree of coverage and deposition of spray liquid for selected air-induction nozzles for leaving the vertical surface.

The results of linear regression for the selected air-induction nozzles are presented in Figures 7–9. Based on the data presented in the figures, it can be concluded that the highest accuracy of the regression model was observed for the leaving vertical surfaces ($R^2 > 0.96$). On the other hand, lower coefficients of determination were obtained for the upper horizontal and approach vertical surfaces (0.84 and 0.86, respectively), meaning a lower quality of models.

## 4. Discussion

The present work analyzed the relationships between two quality indicators of the spraying process separately for standard nozzles and air-induction nozzles. Scientists in their studies highlighted the need to select suitable nozzles and optimal spraying parameters in order to achieve the intended biological effect of spraying [50,51]. This aspect concerns both field, orchard, and horticultural crops, regardless of the equipment used, i.e., ground, hand-held, knapsack, or UAV sprayers. The higher values of the degree of coverage and deposition of liquid were obtained for standard nozzles compared to air-induction nozzles. However, this dependence is more visible in the case of horizontal upper surfaces. This confirms the results of similar research, which were published in the state-of-the-art literature.

The results obtained by Cerruto et al. [45,46] are consistent with the results obtained in this study. The research was conducted in the orchard using an ATR 80 hollow-cone nozzle and four values of liquid pressure, namely 300, 500, 1000 i 1500 kPa. Based on the results, a significant correlation was revealed at the significance level $p < 0.001$ and a regression coefficient of $R^2 = 0.761$ was calculated. Determination coefficients ranging from 0.9949 to 0.9996 were obtained, on the assumption that the values of the mean droplet diameter (VMD) and the coefficient of variation CV are known. Whereas Penido et al. [52] conducted research on the crop of tomatoes. Based on the experimental results, the linear correlation coefficient between the analyzed indicators was calculated ($R = 0.7987$).

Li et al. [53] developed and validated mathematical models to determine the effect of spray performance on droplet size and velocity in the flow field. The authors obtained models of high accuracy ($R^2 > 0.86$). In the work of Cerruto et al. [54], the authors presented a comparison between measured and theoretical distribution for different nozzles. Research on the quality of coverage was the subject of many studies. Biocca et al. [19] showed that the quality of coverage and the number of droplets per unit area were statistically comparable, irrespective of the nozzles used. Holownicki et al. [16] carried out studies in apple orchards using different nozzles and adjuvants. Scientists reported that the liquid coverage of plants during coarse spray at high adjuvant concentrations was comparable to the liquid coverage of plants during fine spray with no added adjuvants. Simao et al. [55] carried out tests using adjuvants regarding the deposition of spray liquid. The researchers stated that similar results of the liquid deposition were obtained using the coarse spray, which means they are less prone to variable spraying parameters. Correlation analysis was employed to validate the Computational Fluid Dynamics model. Computational Fluid Dynamic models have allowed the correct prediction of the general behavior of the fluid in the tank [56–58].

## 5. Conclusions

The analysis of our results provided information on the relationship between the deposition of liquid and the degree of coverage using selected standard and air-induction nozzles. High or very high Pearson correlation coefficients were observed between the deposition of spray liquid and the degree of coverage for both standard and air-induction nozzles. The results are statistically significant ($p < 0.05$). The high values of the determination coefficients $R^2$ indicated a good match of regression models to empirical data. The highest value of the determination coefficient was calculated for standard nozzles on the leaving vertical surfaces ($R^2 = 0.9751$) and the lowest for the air-induction nozzles on the upper horizontal surface ($R^2 = 0.836$). The spray liquid was not found on the bottom horizontal surfaces.

**Author Contributions:** Conceptualization, B.C.; Data curation, B.C.; Formal analysis, B.C.; Investigation, B.C. and T.S.; Methodology, B.C., K.P. and T.S.; Resources, B.C.; Validation, B.C. and K.P.; Visualization, B.C. and K.P.; Writing—original draft, B.C.; Writing—review and editing, K.P. and T.S. All authors have read and agreed to the published version of the manuscript.

**Funding:** The Wrocław University of Environmental and Life Sciences, Łukasiewicz Research Network—Poznań Institute of Technology.

**Institutional Review Board Statement:** Not applicable.

**Informed Consent Statement:** Not applicable.

**Data Availability Statement:** Data are available by contacting the authors.

**Conflicts of Interest:** The authors declare no conflict of interest.

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
