# Peer review of "Correlation and Regression Analysis of Spraying Process Quality Indicators"

_applsci, doi:10.3390/app122312034_

Round 1

Reviewer 1 Report

Interesting article, however, the authors should make changes.

The purpose of the study is not clearly explained. An analysis of two methods for determining the quality of the spray performed was conducted. The correlation coefficients between the results obtained from these methods were determined. The fundamental question is where to apply the obtained results and what can they be used for, both in laboratory research and in agricultural practice? This should be clarified. Otherwise it will be research for research's sake contributing nothing to both science and practice.

The sentence "The analysis of correlation coefficient values or regression models are very common approaches in various fields and disciplines such as medicine [26, 27], biology [30, 31], food technology [32,33], earth sciences [34, 35], economics [36, 37] and agriculture [38- 41]. Therefore, these techniques are employed in this research." contributes nothing, as the information contained therein is generally known. Only the sentence increases the amount of literature cited. 

Was a study done on the uniformity of the distribution of the sprayed liquid from the boom spraying the model plants? If so, please state what the coefficient of variation (CV) was.

I don't understand why the results obtained from two nozzles were combined on one graph. In my opinion, it would be more valuable to test each nozzle separately at different spray pressures or speeds.

With the current juxtaposition, the value of the correlation coefficients could be the result of the nozzle types used and any other combination of nozzles could give different values of the coefficients.

Why were pressures of 200 kPa and 400 kPa selected and 300 kPa omitted?

The figures should indicate which results apply to which nozzles and pressures.

Author Response

Dear Reviewer,

Thank you for your comments and suggests concerning our manuscript entitled “Correlation and regression analysis of spraying process indicators quality” (applsci-2005602). Those comments are all valuable and very helpful for revising and improving our paper, as well as the important guiding significance to our researches. We have studied comments carefully and have made correction which we hope meet with approval. Revised portion are marked in red in the paper. The main corrections in the paper and the responds to the reviewer’s comments are as following.

Point 1: The purpose of the study is not clearly explained. An analysis of two methods for determining the quality of the spray performed was conducted. The correlation coefficients between the results obtained from these methods were determined. The fundamental question is where to apply the obtained results and what can they be used for, both in laboratory research and in agricultural practice? This should be clarified. Otherwise it will be research for research's sake contributing nothing to both science and practice.

Response 1: The answer to the question was included in the discussion: „Biological efficacy depends on the cover and deposition of liquid on plants. The models developed in this study are of high practical use. Based on the knowledge of the degree of coverage (which is faster and easier for determination) the value of deposition of spray liquid can be estimated with high precision. The results of the study may be a reference point for studies on the evaluation of the biological efficacy of the spraying process. Future studies in this area should focus on experiments in field conditions”.

Point 2: The sentence "The analysis of correlation coefficient values or regression models are very common approaches in various fields and disciplines such as medicine [26, 27], biology [30, 31], food technology [32,33], earth sciences [34, 35], economics [36, 37] and agriculture [38- 41]. Therefore, these techniques are employed in this research." contributes nothing, as the information contained therein is generally known. Only the sentence increases the amount of literature cited. 

Response 2: This section is upgraded to present results in range of technique  of plant protection.

Point 3: Was a study done on the uniformity of the distribution of the sprayed liquid from the boom spraying the model plants? If so, please state what the coefficient of variation (CV) was.

Response 3: The uniformity studies were not performed, therefore the results of the CV coefficient of variation were not presented. The flow rate of the liquid was measured before each measurement.

Point 4: I don't understand why the results obtained from two nozzles were combined on one graph. In my opinion, it would be more valuable to test each nozzle separately at different spray pressures or speeds.

Response 4: The research assumption in this study was to determine the relationship between deposition of the liquid and the degree of coverage for various types of nozzles. Therefore, we conducted a correlation and regression analysis with the division into standard and air induction nozzles. Similar correlation and regression relationships were obtained when analyzed each nozzle separately.

Point 5: With the current juxtaposition, the value of the correlation coefficients could be the result of the nozzle types used and any other combination of nozzles could give different values of the coefficients.

Response 5: The research results presented in the study are preliminary research which will be continued. The results of experiments with the use of various types of nozzles as well as technical and technological factors will be presented in further publications.

Point 6: Why were pressures of 200 kPa and 400 kPa selected and 300 kPa omitted?

Response 6: In accordance with the research assumption, two pressure levels were selected for the experiments. Each successive factor would cause a multiplication of the combinations to be analyzed.

Point 7: The figures should indicate which results apply to which nozzles and pressures.

Response 7: The figures was corrected, according with suggest of Reviewer.

Reviewer 2 Report

Dear Authors,

The manuscript entitled “Correlation and regression analysis of indicators of quality of spraying process” was revised. As a result, it was seen that there were many grammatical errors in the text and these errors should be corrected before publication. I made some corrections and suggestions in the review report. Please find the related report as attached file. 

Best ragards

Author Response

Dear Reviewer,

Thank you for your comments and suggests concerning our manuscript entitled “Correlation and regression analysis of spraying process indicators quality” (applsci-2005602). Those comments are all valuable and very helpful for revising and improving our paper, as well as the important guiding significance to our researches. We have studied comments carefully and have made correction which we hope meet with approval. Revised portion are marked in red in the paper. The main corrections in the paper and the responds to the reviewer’s comments are as following.

Point 1: Title

It seems more appropriate for title, this is only a suggestion.

“Correlation and regression analysis of spraying process quality indicators”

Abstract

The "of" preposition is overused, which takes the meaning of the sentence away from its academic nature. On the other hand, there are some grammatical errors, for example, “The study present+s the results of the correlation……………”

In general, this part should be improved.

Keywords

Keywords should be at a level to identify the current article when searching for topics similar to the article. The ones given here are of the type that will be encountered when scanning for other purposes. For example "spraying" or “degree of coverage”. For this reason, keywords should be reconsidered and should qualify the article. Also, keywords do not need to include words in the title.

Introduction

# The first paragraph, the more suitable “………..is the use of chemicals in plant protection…..” Ä°nsted of “…………….is chemical plant protection…” ……… “However, other limiting factors such as drift of spraying……can occur during spraying…..”

There are many parts of the manuscript like this that need to be corrected. Therefore, the entire manuscript should be reconsidered.

Response 1: Linguistic correction has been done by a native speaker.

Point 2: Materials and Methods

The method is detailed, but the applications should be simpler and more understandable in the explanation.

Response 2: The Reviewer's suggestion was taken into account.

Results

OK

Point 3: Discussion

We analyzed the relationships between two quality indicators of the spraying process separately for standard nozzles and air induction nozzles in the present work. The many scientists in their studies highlighted the need to select suitable nozzles and optimal spraying parameters in order to achieve the intended biological effect of spraying[need references]. This section is too short and it needs to expand. Especially, the data obtained should be discussed further by emphasizing the purpose of the study.

Response 3: The Reviewer's suggestion was taken into account.

Point 4: Conclusions

It would be better if the conclusion part was given as a single paragraph.

Response 4: The Reviewer's suggestion was taken into account.

Point 5: Author Contributions: Authors' abbreviations are B.C.; K.P. and T.S X.X; Y.Y; Z.Z…. ??? References The references should be arranged one by one according to the spelling rules of the journal (https://www.mdpi.com/authors/layout).

Such as, the journal names should be as italic………

The years should be write bold……

Response 5: We have modified according the requirements by the journal.

Reviewer 3 Report

The author studied the correlation between the liquid deposition amount and the coverage degree through experiments, and finally concluded that the deposition amount is linearly related to the coverage degree. The overall design of the article is reasonable, the experimental process is complete, and on the whole, it is relatively standard, but there are still the following problems:

1. The author should compare and modify the text format of the article and the reference format in strict accordance with the requirements of the journal;

2. Improve the clarity of Figure 2;

3. Under the condition of the same atomized particle size, the target with high coverage has a large liquid deposition, which is obvious. What is the significance of the author's experimental verification of this conclusion?

4. The introduction of the author does not reflect the importance of the research. First, there is no clear explanation of the research status of the paper. In addition, the index literature in this part is not closely related to the research focus of the article.

Author Response

Response to Reviewer 3 Comments

Dear Reviewer,

Thank you for your comments and suggests concerning our manuscript entitled “Correlation and regression analysis of spraying process indicators quality” (applsci-2005602). Those comments are all valuable and very helpful for revising and improving our paper, as well as the important guiding significance to our researches. We have studied comments carefully and have made correction which we hope meet with approval. Revised portion are marked in red in the paper. The main corrections in the paper and the responds to the reviewer’s comments are as following.

Point 1: The author should compare and modify the text format of the article and the reference format in strict accordance with the requirements of the journal;

Response 1: We have modified the text format complying with the requirements by the journal.

Point 2: Improve the clarity of Figure 2;

Response 2: Figure 2 was corrected.

Point 3: Under the condition of the same atomized particle size, the target with high coverage has a large liquid deposition, which is obvious. What is the significance of the author's experimental verification of this conclusion?

Response 3: The aim of the experiments was to develop models of correlation and regression between the indicators of the quality of the spraying treatment. The degree of coverage is an easier and faster indicator to determine. Therefore, based on the knowledge of the coverage degree value, the spray liquid application value can be estimated, and then the biological effectiveness of the spraying treatment can be predicted. The developed models have a practical aspect.

Point 4: The introduction of the author does not reflect the importance of the research. First, there is no clear explanation of the research status of the paper. In addition, the index literature in this part is not closely related to the research focus of the article.

Response 4: The introduction has been supplemented with information about correlation and regression in technique of plant protection.

Reviewer 4 Report

Reviewer Comments V1:

Why are the line numbers missing? They are very useful for reviewing a scientific paper.

Original text: arial narrow

Revision text: Calibri

Abstract:

“The results of these studies showed strong and very strong Pearson’s correlation coefficient between analyzed indicators”

I think it would be better for the reader to know the correlation figure than "strong" and "very strong", and specifically "between which" variables. Linear regression? Between which variables? The authors should, in the abstract, "engage the reader" with concrete data.

1.- Introduction

“One of the primary practices of agricultural production is chemical plant protection because it increases the control of pests, and ensures high quantity and quality of yield”

The first, and important, assertion made by the authors in the introduction must be based on "proven facts" and should therefore be accompanied by a bibliographic citation to support this assertion.

effectiveness. [5, 6]

The full stop at the end of the word and before the bracket is unnecessary.

The impact of technical and technological factors on this indicator was described based on the results of research [26, 27].

Isn't this more appropriate for the Material and Methods section?

2. Materials and Methods

The assessment of the coverage degree was conducted in the graphic program Adobe Photoshop.

This is the first question I ask about how the authors are doing things but... it seems to me that there are few explanations and they are not clear enough for others to replicate the same work.

In addition, in previous paragraphs, they do not provide the technical characteristics of the devices they use, which could be provided in supplementary material.

The number of droplets was recorded using computer soft-ware.

Which one? I insist, the authors should provide all the information, as complete as possible, in case someone wants to replicate their experiment or discuss the results of a part of the procedure.

Figure 3

When you hover the mouse over the numbers indicating elements in Figure 3, you will see signs with oriental lettering that... are "not understandable". I suggest removing these signs when hovering over the numbers.

3. Results

Table 3

Firstly, the table has to be "self-explanatory", so the meaning of the abbreviations in the first column should appear in the table heading.

Secondly, we would be more interested in knowing a centrality parameter (the sample mean) and a dispersion parameter (the sample quasi-variance or sample standard deviation) than the values to compare with the F-table.

Thirdly, if we were to add something from the analysis of variance, we would be more interested in knowing the Sum of Squares of each of the variables with respect to that of the error.

Fourthly, adding a column with all values equal: P-value, does not make any sense; it can be commented in the epigraph that the p-value is < 0.001 and is thus well defined.

It should also be explained in the epigraph that "R" is the correlation coefficient, which, in turn, may (or may not) also be significant, whose p-value would be a real coincidence if it coincided with that of the ANOVA.

This table must be "mandatorily" improved by the authors.

4. Discussion

In addition to the few papers cited by the authors, there is more literature on some of the parameters they analyse, a few of which I mention as examples:

Badules, J., Vidal, M., Boné, A., Gil, E., & García-Ramos, F. J. (2019). CFD Models as a Tool to Analyze the Performance of the Hydraulic Agitation System of an Air-Assisted Sprayer. Agronomy, 9(11), 769. https://doi.org/10.3390/agronomy9110769

García-Ramos, F. J., Malón, H., Aguirre, A. J., Boné, A., Puyuelo, J., & Vidal, M. (2015). Validation of a CFD Model by Using 3D Sonic Anemometers to Analyse the Air Velocity Generated by an Air-Assisted Sprayer Equipped with Two Axial Fans. Sensors, 15(2), 2399-2418. https://www.mdpi.com/1424-8220/15/2/2399

García-Ramos, F. J., Vidal, M., Boné, A., Malón, H., & Aguirre, A. J. (2012). Analysis of the Air Flow Generated by an Air-Assisted Sprayer Equipped with Two Axial Fans Using a 3D Sonic Anemometer. Sensors, 12(6), 7598-7613. https://www.mdpi.com/1424-8220/12/6/7598

In general, the discussion seems to me "very weak" and furthermore, it does not discuss the values obtained: correlation, Coefficient of Determination. It wouldn't hurt to compare one type of nozzles against another... etc. Poor, very poor. I think that the results shown by the authors can be discussed much more.

5. Conclusions

The higher values of the degree of coverage and deposition of liquid were obtained for standard nozzles compared to air induction nozzles.

That... where have the authors shown it? The results of the paper (at least what I have reviewed) do not show that. It was one of my earlier recommendations. Show it!

This confirms the results of similar research which has been published in the state of art literature

This statement (confirmation) is not required in the conclusion section, but is required in the discussion.

High or very high Pearson’s correlation coefficients were observed between deposition of spray liquid and degree of coverage for both, standard and air induction nozzles. The results are statistical significant (p < 0.05).

What is of real interest in the conclusions is the value of the correlation coefficient, or at least the range obtained. The p-value is superfluous, it is enough to say that they were significant.

“statistically” by “statistical”

Overall, I think the paper is interesting. The essay is well thought out (but poorly explained). The results are relevant but... very brief and important information that is of interest to readers and researchers is left out. The discussion should be expanded in a way that discusses the parameters obtained (and some of their causes) and the conclusions should be nuanced. I encourage the authors to improve all these aspects and they will be able to provide relevant information to the readers of this journal.

Author Response

Response to Reviewer 4 Comments

Dear Reviewer,

Thank you for your comments and suggests concerning our manuscript entitled “Correlation and regression analysis of spraying process indicators quality” (applsci-2005602). Those comments are all valuable and very helpful for revising and improving our paper, as well as the important guiding significance to our researches. We have studied comments carefully and have made correction which we hope meet with approval. Revised portion are marked in red in the paper. The main corrections in the paper and the responds to the reviewer’s comments are as following.

Point 1: Why are the line numbers missing? They are very useful for reviewing a scientific paper.

Original text: arial narrow

Revision text: Calibri

Response 1: The text format has been modified.

Point 2: Abstract:

“The results of these studies showed strong and very strong Pearson’s correlation coefficient between analyzed indicators”

I think it would be better for the reader to know the correlation figure than "strong" and "very strong", and specifically "between which" variables. Linear regression? Between which variables? The authors should, in the abstract, "engage the reader" with concrete data.

Response 2: The Reviewer's suggestion was taken into account.

Point 3: 1.- Introduction

“One of the primary practices of agricultural production is chemical plant protection because it increases the control of pests, and ensures high quantity and quality of yield”

The first, and important, assertion made by the authors in the introduction must be based on "proven facts" and should therefore be accompanied by a bibliographic citation to support this assertion.

effectiveness. [5, 6]

The full stop at the end of the word and before the bracket is unnecessary.

Response 3: The Reviewer's comments were taken into account.

Point 4: The impact of technical and technological factors on this indicator was described based on the results of research [26, 27].

Isn't this more appropriate for the Material and Methods section?

Response 4: All the technical and technological factors were described the introduction. The factors that should be considered during spraying. Therefore, this information should remain in the introductory section.

Point 5: 2. Materials and Methods

The assessment of the coverage degree was conducted in the graphic program Adobe Photoshop.

This is the first question I ask about how the authors are doing things but... it seems to me that there are few explanations and they are not clear enough for others to replicate the same work.

In addition, in previous paragraphs, they do not provide the technical characteristics of the devices they use, which could be provided in supplementary material.

The number of droplets was recorded using computer soft-ware.

Which one? I insist, the authors should provide all the information, as complete as possible, in case someone wants to replicate their experiment or discuss the results of a part of the procedure.

Response 5: The Reviewer's suggestion was taken into account.

Point 6: Figure 3

When you hover the mouse over the numbers indicating elements in Figure 3, you will see signs with oriental lettering that... are "not understandable". I suggest removing these signs when hovering over the numbers.

Response 6: Figure 3 was paste as an image.

Point 7: 3. Results

Table 3

Firstly, the table has to be "self-explanatory", so the meaning of the abbreviations in the first column should appear in the table heading.

Secondly, we would be more interested in knowing a centrality parameter (the sample mean) and a dispersion parameter (the sample quasi-variance or sample standard deviation) than the values to compare with the F-table.

Thirdly, if we were to add something from the analysis of variance, we would be more interested in knowing the Sum of Squares of each of the variables with respect to that of the error.

Fourthly, adding a column with all values equal: P-value, does not make any sense; it can be commented in the epigraph that the p-value is < 0.001 and is thus well defined.

It should also be explained in the epigraph that "R" is the correlation coefficient, which, in turn, may (or may not) also be significant, whose p-value would be a real coincidence if it coincided with that of the ANOVA.

This table must be "mandatorily" improved by the authors.

Response 7: The Reviewer's suggestion was taken into account.

Point 8: 4. Discussion

In addition to the few papers cited by the authors, there is more literature on some of the parameters they analyse, a few of which I mention as examples:

Badules, J., Vidal, M., Boné, A., Gil, E., & García-Ramos, F. J. (2019). CFD Models as a Tool to Analyze the Performance of the Hydraulic Agitation System of an Air-Assisted Sprayer. Agronomy, 9(11), 769. https://doi.org/10.3390/agronomy9110769

García-Ramos, F. J., Malón, H., Aguirre, A. J., Boné, A., Puyuelo, J., & Vidal, M. (2015). Validation of a CFD Model by Using 3D Sonic Anemometers to Analyse the Air Velocity Generated by an Air-Assisted Sprayer Equipped with Two Axial Fans. Sensors, 15(2), 2399-2418. https://www.mdpi.com/1424-8220/15/2/2399

García-Ramos, F. J., Vidal, M., Boné, A., Malón, H., & Aguirre, A. J. (2012). Analysis of the Air Flow Generated by an Air-Assisted Sprayer Equipped with Two Axial Fans Using a 3D Sonic Anemometer. Sensors, 12(6), 7598-7613. https://www.mdpi.com/1424-8220/12/6/7598

In general, the discussion seems to me "very weak" and furthermore, it does not discuss the values obtained: correlation, Coefficient of Determination. It wouldn't hurt to compare one type of nozzles against another... etc. Poor, very poor. I think that the results shown by the authors can be discussed much more.

Response 8: The Reviewer's suggestion was taken into account.

Point 9: 5. Conclusions

The higher values of the degree of coverage and deposition of liquid were obtained for standard nozzles compared to air induction nozzles.

That... where have the authors shown it? The results of the paper (at least what I have reviewed) do not show that. It was one of my earlier recommendations. Show it!

This confirms the results of similar research which has been published in the state of art literature

This statement (confirmation) is not required in the conclusion section, but is required in the discussion.

High or very high Pearson’s correlation coefficients were observed between deposition of spray liquid and degree of coverage for both, standard and air induction nozzles. The results are statistical significant (p < 0.05).

What is of real interest in the conclusions is the value of the correlation coefficient, or at least the range obtained. The p-value is superfluous, it is enough to say that they were significant.

“statistically” by “statistical”

Response 9: The Reviewer's suggestion was taken into account.

Round 2

Reviewer 1 Report

Thank you for considering the comments and providing explanations.

However, I have a few more comments.

The explanation of why such a study aim was adopted should be in the introduction and not in the discussion.

In the answer to item 3 there is information about the test of liquid flow rate. This information should be included in the methodology, for example, in Table 1.

It is still not explained why the pressures of 200 kPa and 400 kPa were adopted. The explanation does not inform why such a research assumption was made. This information should be included in the M&M section.

Author Response

Dear Reviewer,

Thank you for your comments and suggests.

The main corrections in the paper and the responds to the reviewer’s comments are as following.

Point 1: The explanation of why such a study aim was adopted should be in the introduction and not in the discussion.

Response 1: The reviewer's suggestion was taken into account.

Point 2: In the answer to item 3 there is information about the test of liquid flow rate. This information should be included in the methodology, for example, in Table 1.

Response 2: The reviewer's suggestion was taken into account.

Point 3: It is still not explained why the pressures of 200 kPa and 400 kPa were adopted. The explanation does not inform why such a research assumption was made. This information should be included in the M&M section.

Response 3: The highest and lowest value of the liquid pressure due to the nozzles used – was added in section 2.5.

Reviewer 2 Report

Dear Authors,

All the corrections I have suggested were done . The article can be published in its current form after a final check by the authors.

Kind regards

Author Response

Dear Reviewer,

Thank you for your comments and suggests.

Reviewer 3 Report

It can be clearly seen that the author has made some modifications to the paper, but there are still some shortcomings.

1. The introduction of the paper is confused, not logical, and does not highlight the significance of the article

2. The picture quality is still poor. What is the difference between the two pictures in Figure 2? The annotation of the first picture is very confused. In addition, the legend in Figure 4-9 is obviously pasted, resulting in very low clarity, which should be taken seriously by the author.

Author Response

Dear Reviewer,

Thank you for your comments and suggests.

The main corrections in the paper and the responds to the reviewer’s comments are as following.

It can be clearly seen that the author has made some modifications to the paper, but there are still some shortcomings.

Point 1: The introduction of the paper is confused, not logical, and does not highlight the significance of the article.

Response 1: The reviewer's suggestion was taken into account.

Point 2: The picture quality is still poor. What is the difference between the two pictures in Figure 2? The annotation of the first picture is very confused. In addition, the legend in Figure 4-9 is obviously pasted, resulting in very low clarity, which should be taken seriously by the author.

Response 2: Picture in Figure 2 was changed. Maybe the Reviewer turned on the track changes mode, therefore two figures No. 2 were visible. Figures 4-9 were changed.

Reviewer 4 Report

L17-19: “In addition, high coefficient of determination (R2) values (>0.85) were obtained for the linear regression (between which variables???), which seems to indicate the good fit of the proposed model.”

L241-243: Correlation analysis could be used to validate the Computational Fluid Dynamics model. Computational Fluid Dynamics models allow the overall behaviour of the fluid in the tank to be correctly predicted [56-58].

Author Response

Dear Reviewer,

Thank you for your comments and suggests.

Point 1: L17-19: “In addition, high coefficient of determination (R2) values (>0.85) were obtained for the linear regression (between which variables???), which seems to indicate the good fit of the proposed model.”

Response 1: The reviewer's suggestion was taken into account.